# The Adaptation of the Communicative Effectiveness Index (CETI) into Greek: A Reliability and Validity Study

**DOI:** 10.3390/brainsci14070689

**Published:** 2024-07-10

**Authors:** Marina Charalambous, Phivos Phylactou, Eleftheria Antoniou, Maria Christodoulou, Maria Kambanaros

**Affiliations:** 1Department of Rehabilitation Sciences, Cyprus University of Technology, Limassol 3041, Cyprus; marina.charalambous@cut.ac.cy (M.C.); eg.antonniou@edu.cut.ac.cy (E.A.); mariaqcc.christodoul@edu.cut.ac.cy (M.C.); 2School of Physical Therapy, University of Western Ontario, London, ON N6G 3K7, Canada; pphylact@uwo.ca; 3The Gray Centre for Mobility and Activity, Parkwood Institute, London, ON N6A 4V2, Canada; 4The Brain and Neurorehabilitation Lab, Department of Rehabilitation Sciences, Cyprus University of Technology, Limassol 3041, Cyprus

**Keywords:** people with aphasia, stroke, functional communication, tool adaptation, CETI-GR

## Abstract

The Communicative Effectiveness Index (CETI) is an informant rating scale designed to measure changes in functional communication in people with aphasia (PWA) from the carer’s perspective. It offers a comprehensive view of aphasia’s impact on everyday communication situations, aiding clinicians in designing personalized intervention plans. The aim of this study was to translate and adapt the CETI into Greek (CETI-GR) and validate its psychometric properties. The CETI-GR was translated into Greek using back-translation. A pilot and a content validity study ensured its acceptability. The study involved 30 people with aphasia and 30 carers. The CETI-GR’s psychometric properties were evaluated, including internal consistency, test–retest reliability, inter-rater reliability, and validity measures. The CETI-GR demonstrated excellent internal consistency (Cronbach’s α > 0.95) and excellent inter-rater reliability (ICC ≥ 0.93). Excellent consistency was revealed when testing the CETI responses given only by PWA (α = 0.91) versus their carers (α = 0.97). Test–retest reliability was high (ICC = 0.88). Significant correlations between the CETI-GR and measures of language severity, functional communication, and quality of life supported convergent validity. The CETI-GR is a reliable tool for assessing functional communication in chronic aphasia. Its Greek adaptation enhances aphasia rehabilitation, enabling person-centered care and improving the quality of life for people with aphasia and carers.

## 1. Introduction

Communication is a fundamental aspect of human interaction, serving as the bedrock for the expression of thoughts, emotions, and intentions. However, for people living with chronic aphasia, a communication impairment often resulting from stroke, everyday communication can seem a formidable challenge [1]. Aphasia affects the ability to understand, produce, and process language, causing difficulties with functional communication for people living with aphasia such as conversing with others, expressing needs and desires, and maintaining personal relationships and friendships [2]. Moreover, persisting difficulties in communicating effectively may contribute to isolation and mood disorders [3]. For people with aphasia (PWA) and their carers, successful communication is related to improved relationships, social engagement, and functional ability, which are also associated with a better quality of life [4,5].

### 1.1. Evidence-Based Functional Communication Assessment Measures

While traditional standardized aphasia assessment tools focus on the linguistic abilities of the person with aphasia, functional communication tools assess the individual’s ability to effectively communicate in real-life situations [6]. Recent evidence suggests that functional communication assessment tools are essential for comprehensive aphasia management and rehabilitation [7]. The Research Outcome Measurement in Aphasia Consensus Statement (ROMA-COS) as outlined by Wallace et al. [8], recommends the implementation of two evidence-based assessment tools for evaluating functional communication: the Scenario Test [9] and the Communicative Effectiveness Index (CETI) [10]. Both the Scenario Test [9] and the CETI [10] assess functional communication abilities in people with chronic aphasia but differ in approach and focus. The aim of the Scenario Test is to assess the functional communication abilities of the person with aphasia specifically, by simulating real-life scenarios [9], whereas the CETI assesses the functional communication abilities of PWA from the perspectives of their carers [10]. Each tool offers unique insights into different aspects of functional communication and may be used complementarily to provide a comprehensive assessment of communication abilities.

### 1.2. The Communication Effectiveness Index (CETI)

The CETI was originally developed in English [10], to offer a structured approach for assessing the functional communication abilities of PWA living in Canada from the perspectives of their carers [9]. The CETI is a short questionnaire, representing 16 different everyday communication situations (items), and taps into verbal communication abilities (10 items) and non-verbal (6 items) communication skills [8,10]. Each item is rated out of 100 points. Carers are asked to rate the effectiveness of the communication of the person with aphasia across the various situations, including social interactions and participation in meaningful conversations. The rating for each situation is converted into a score by laying a template marked with 1 mm divisions over the 10-cm visual analog scale and reading off a value between 1 and 100. The total CETI score is converted to a 100-point maximum by dividing the sum of the individual situation ratings by the total number of situations. A high score indicates good performance in functional communication and a low score, poor performance [7,10]. The CETI provides a broad overview of communication effectiveness of PWA in their everyday lives, capturing the impact of aphasia on communication from the perspective of those who interact with them daily [8,10].

### 1.3. Other Language Adaptations of the CETI

The CETI has been proven to be a useful tool for evaluating functional communication in PWA [8]. However, it has limited applicability for people with chronic aphasia in non-English-speaking settings. Since its original publication in English by Lomas et al. [10], only three adaptations have been published chronologically as follows: The South African version by Penn et al. [11];The Danish version by Pedersen et al. [12];The Italian version by Moretta et al. [13].

See Table 1 for a description of the participants, the psychometric properties of the CETI, and the comparison tools used from each of the above studies.

### 1.4. The Importance of the Greek Validation of the CETI 

It is important to recognize that language and culture can have a significant impact on how functional communication assessment tools are developed. This emphasizes the need for adapting assessment tools to meet the needs of diverse linguistic communities. For example, in Cyprus, where this study was completed, Standard Modern Greek is the variety used in formal oral and written communication, and Cypriot Greek, a dialect of Standard Modern Greek, is the mother tongue of Greek Cypriots and is used in informal interactions [25]. Since cultural norms differ from those in English-speaking countries, it is crucial to have culturally and linguistically adapted measures to evaluate the functional communication skills and effectiveness of Greek-speaking individuals with persisting communication difficulties because of aphasia.

While the Greek version of the Scenario Test [6] is currently available, clinicians should not disregard the significance of incorporating in their assessment the perspective of the carer, as their viewpoint is essential for numerous reasons. Firstly, assessing functional communication from the perspective of the carer provides a holistic understanding of the daily impact of living with aphasia for both the individual and the carer [26]. By soliciting the carer’s perspective, clinicians gain valuable insights into the specific communication challenges faced by PWA in real-life situations, which standardized language assessments or self-report measures may not fully capture [10]. Secondly, involving carers in the assessment process enhances collaborative goal setting and treatment planning. Carers provide a unique perspective into the communication abilities of PWA, encompassing linguistic, pragmatic, social, and functional aspects [10,13]. This holistic perspective is essential for developing tailored intervention plans. Additionally, involving carers in the assessment process ensures that intervention plans are aligned with the priorities and preferences of both the individual and their carers [27]. Further, carers play a crucial role in monitoring communication progress over time, providing feedback about the effectiveness of the interventions implemented, which allows clinicians to make the necessary adjustments to optimize outcomes [13]. Finally, validating the CETI in Greek opens up opportunities for research and collaboration in aphasia rehabilitation and treatment, contributing to the broader knowledge base in aphasiology and the development of best practices for addressing the communication needs of Greek-speaking PWA in the chronic phase of stroke. 

### 1.5. Foundational Theoretical Frameworks 

The adaptation of the CETI into Greek was informed by key theoretical frameworks, such as the Social Model of Disability [28] and person-centered care [29], to ensure that the CETI not only measures communication effectiveness but does so in a way that is inclusive and respectful of the individuals using it, while addressing societal factors and promoting person-centered approaches. Specifically, the Social Model of Disability refers to the recognition of communication barriers beyond language impairments [30], aiming to address societal and environmental factors hindering PWA from engaging in social interactions [31]. Assessing the communication skills of people with aphasia in real-life situations and environments can provide valuable insights into specific communication barriers and facilitators. This information can ultimately guide intervention strategies and promote social inclusion and participation.

Moreover, the person-centered care framework [29] was implemented in this adaptation by following a Patient and Public Involvement (PPI) approach and actively involving a research (PPI) partner with aphasia [32]. The PPI partner was involved during the adaptation phase of the CETI as described in the Section 2. A content validity, study codesigned with the PPI partner, was completed with the involvement of various stakeholders such as people with aphasia and stroke, family members, and speech and language therapists. The inclusion of these stakeholders during the adaptation and validation phase ensured that the CETI was tailored to their needs, adhering to the principles of person-centered care [33]. By seeking input, feedback, and perspectives from all stakeholders, the adaptation of the CETI into Greek aimed to authentically reflect the diverse needs, preferences, and experiences of the target population [34]. This inclusive and participatory process ensured that the adapted CETI is not only culturally and linguistically appropriate but also relevant to PWA, thereby upholding the core tenets of person-centered care [35].

Drawing upon the principles of cross-cultural adaptation, psychometrics, and rigorous validation procedures, we sought to ensure that the Greek version of the CETI is a reliable, valid, and sensitive measure able to inform clinical practice, research, and policy efforts to improve the communication outcomes and quality of life of people with chronic aphasia. 

### 1.6. Aim

This study aimed to adapt the CETI into Greek (CETI-GR) and assess its reliability and validity.

## 2. Materials and Methods

This study examines the validation of the Greek version of the 16 items of the CETI based on the original English version by Lomas et al. [10]. The CETI was translated and adapted into Standard Modern Greek by the authors. Standard Modern Greek is one of the two official languages of the Republic of Cyprus, the other being Turkish [25]. The questionnaire was translated into Standard Modern Greek by the first author (MCha) and was back-translated into English by the senior author (MK), a balanced English–Greek bilingual, resulting in a very high correspondence to the original source (90% agreement). Back-translation ensures accuracy and cultural relevance by identifying discrepancies between the original and translated versions. Despite involving Greek-speaking participants, back-translation into English maintains the translation’s integrity, ensuring the content’s meaning remains consistent. Having an author fluent in both languages perform the back-translation enhances accuracy due to their deep understanding of the subject matter and linguistic nuances. The first author (MCha) and the PPI partner with aphasia engaged in face-to-face discussions until a consensus was reached for all 16 items (see Appendix A). During this meeting, the 16 translated items were reviewed and revised by the PPI partner. For example, the PPI partner suggested adapting the word “strangers” in item 15 to “άγνωστους”/aγnostus/, as the direct translation for “ξένοι” (xenoi) implies “people I might know but are not related to me” in the Cypriot dialect. 

A pilot study was conducted to check for the acceptability of the Greek version of the 16 items and the time taken for administration and scoring. The pilot sample group consisted of 5 people with chronic stroke (greater than 6 months post stroke) and 5 carers, recruited from the Cyprus Stroke Association registry. The CETI-GR was administered to all participants in one session with an average administration time of 10–15 min depending on their language skills. The pilot study resulted in a wide range of scores (ranging between 18 and 98 out of 100).

### 2.1. Design

A cross-sectional study was carried out to evaluate the psychometric properties of the Greek version of the CETI. 

### 2.2. Participation Criteria

All participants included in the study met the pre-established inclusion and exclusion criteria. The inclusion criteria for PWA were (1) to be native Greek speakers, (2) to be adults (18+), (3) to have suffered a stroke as confirmed by neuroimaging (computed tomography or magnetic resonance imaging), (4) to be in the chronic phase of stroke (>6 months), and (5) to present with aphasia, of any type and severity, as diagnosed by speech and language therapy service providers during rehabilitation. 

The exclusion criteria for PWA were as follows: (1) to present with hearing and visual impairments that interfere with the completion of the study, (2) to have an additional diagnosis of a degenerative disease or traumatic brain injury, and/or (3) to have a confirmed diagnosis of a psychiatric disorder. The above criteria were established to ensure that the language or functional communication deficits were a result of stroke or aphasia rather than cognitive or sensory impairments. Medical history regarding hearing and vision was determined by observation, self-report, and reports from the carer during the screening interview. 

The inclusion criteria for carers were to be (1) Greek speaking and (2) formal/professional carers working as personal assistants at home, as carers at private centers, or as informal carers (e.g., the person’s spouse or another family member). Carers who had contact less than 3 times per week with a person with aphasia were excluded from the study [10].

### 2.3. Recruitment 

The recruitment phase took place between January 2024 and April 2024. Participants were recruited from all districts of the Republic of Cyprus. Recruitment sources were the Melathron Agoniston EOKA Neurorehabilitation Center, the Neurorehabilitation Center in Limassol, the Rehabilitation Clinic of the Cyprus University of Technology in Limassol, the Registry of the Cyprus Stroke Association, the “Sokratio” Melathron Evgirias residential care home and “Vasiliada” retirement home located in Limassol, the “Eden” Rehabilitation Center located in Larnaca, the “Melathron Agapis” care home in Nicosia, and from private speech and language therapy practices in Nicosia and Paphos. 

### 2.4. Sample Size

This study included 60 participants. The sample size was not determined a priori, but instead relied on convenience sampling from willing participants referred from the recruitment sources. Nonetheless, our sample size is considered sufficient for the scope of the study compared to previous sample sizes for the validation of the CETI based on the country’s population and incidence of stroke per year (see Table 2). The data in Table 2 were extracted from the study of Ranganai and Matizirofa [36] for South Africa, the study of Krueger et al. [37] for Canada, and the Burden of Stroke report in Europe [38] for Demark, Italy, and Cyprus for the current study. Further, to explore potential Type I and Type II errors, we conducted post-hoc power analyses to estimate the achieved power of our study. In detail, we conducted two power analyses: one based on internal consistency and a second one based on validity. For the consistency power analysis, we estimated that we achieved a power of β > 99%, considering CETI-GR’s 16 items, an overall consistency of Cronbach’s α = 0.95 (see Section 3.2.1), a minimum accepted Cronbach’s α = 0.8, an α = 0.05, and our sample size of *n* = 60. The validity power analysis was based on the results from the correlational analysis between CETI-GR and AIQ-21-GR (see Section 3.3), considering that this was the lowest correlation calculated. As such, we calculated that we achieved a power of at least β = 96%, with a correlation of ρ = |0.46|, an α = 0.05, and our sample size of *n* = 60.

### 2.5. Participants

Thirty PWA and their respective carers were included in the study. In sum, 60 participants (*n* = 60) took part in the study investigating the psychometric properties of the CETI-GR. Of the 30 participants with stroke-induced aphasia, 14 (47%) were female and 16 were male (see Table 3). Details on the demographics, including type of stroke, localization of the lesion, level of education, etc., are presented in Table 3. Of the 30 carers, 23 (77%) were female (see Table 3). More than 60% of the carers were informal (e.g., the person’s spouse or another family member), while the remaining 40% were formal/professional carers working in people’s homes or in private centers. The age of the participants with aphasia ranged between 36 and 89 years old with a mean age of 67.67 (*sd* = 10.71), and the age of the carers ranged between 26 and 78 years old with a mean age of 47.4 years (*sd* = 16.33). The time post stroke ranged from 6 to 180 months (15 years) with a mean of 44.07 (*sd* = 48.81) months, indicating that all participants were in the chronic phase post stroke. Participant demographics are reported in Table 3.

### 2.6. Data Collection and Procedures 

For the validation of the CETI-GR, ethical approval was obtained from the Cyprus National Bioethics Committee (EEBK ΕΠ 2024.01.109). The tests and questionnaires were administrated either at the participant’s home, in private clinics or at speech and language therapy offices. Testing was conducted by two qualified Greek-speaking speech and language therapists (EA and MC), who received training from the first author (MCha). The first author (MCha) is a senior speech and language therapist working in aphasia assessment and rehabilitation. Informed consent was documented using a written, signed, and dated informed consent form and personal information was obtained from each participant before the beginning of the study. 

The study protocol was completed in three sessions. The first session had a duration of approximately two hours, whereas the second and third sessions were of a 15 min duration each. During the first session, four measures were administered to participating PWA including the Aphasia Severity Rating Scale (ASRS) from the Greek Boston Diagnostic Aphasia Examination [39], the Hospital Anxiety and Depression Scale (HADS) [40] to assess mood disorders, the CETI and the Greek version of the Scenario Test [6] to assess functional communication, and the Greek version of the Aphasia Impact Questionnaire-21 (AIQ-21-GR) to examine the impact of aphasia on the quality of life of PWA [41]. The two speech and language therapists (EA and MC) initially assessed the person with aphasia on all measures. After that, the respective carers completed the CETI-GR and AIQ-21-GR. Carers participated in one-on-one meetings with speech and language therapists (EA and MC), who explained the test’s scope, provided clear scoring guidelines, and addressed any queries about the 16 daily situations to ensure consistent understanding. These steps ensured the assessments were comparable and reliable, regardless of whether the caregivers were formal or informal. The measures used in this study are presented in detail below.

### 2.7. Measures 

A selection of the measures related to validating the CETI-GR included tools that are validated in Greek and tapped into language abilities, functional communication, depression and anxiety, and the impact of aphasia. These were as follows:The Aphasia Severity Rating Scale (ASRS) from the Greek adaptation of the Boston Diagnostic Aphasia Examination Short Form (BDAE-SF) [39]. The ASRS is a rating scale used to measure aphasia severity. This scale was used to evaluate the severity of the observed language and communication difficulties of the participants with aphasia. This included (1) a 10 min semi-structured interview about their previous employment, their stroke story, and basic demographic information and (2) a description of the “Cookie Theft” picture. Aphasia severity was evaluated based on the fluency and intelligibility of the spoken output. The scores of the ASRS ranged from 0 to 5, with 0 indicating very severe non-fluent aphasia and 5, very mild aphasia predominantly characterized by naming difficulties [39].The standardized Greek version of the Hospital Anxiety and Depression Scale (HADS-GR) [40]. The HADS-GR evaluates potential depression and anxiety in people with medical conditions. The HADS-GR is a self-report rating scale with 14 items each rated on a 4-point Likert scale, ranging between 0 and 3. The anxiety and depression subscales contain 7 items each. The total score is calculated by the sum of the 7 items for each subscale. A score of 0–7 indicates no depression (or anxiety, respectively), 8–10 indicates abnormal borderline, and 11–21 indicates severe depression or anxiety [40].The Greek version of the CETI. The CETI is a 16-item questionnaire completed by the carers of people with chronic aphasia. The CETI assesses both verbal (10 items) and non-verbal (6 items) communication skills in 16 different daily situations [10]. Each statement was presented to the respondents using a visual analog scale represented by a horizontal line of 100 mm. Zero means “not able at all” and 100 “as able as before”. Pedersen et al. [12] found that raters often placed their crosses midway on the visual analog scale lines when scoring 0 or 100 on the Danish CETI. Therefore, to improve precision in the Greek version, smiling faces were added to signify key indicators from “not able at all” to “as able as before”, assisting raters in marking the scale more accurately (see Figure 1).For each statement, the carer and the person with aphasia had to mark their response on the visual analog scale with a pencil. Each answer was rated from 0 to 100 and the total score was calculated by dividing the sum of the individual situation ratings by the total number of situations [10]. Lower scores show lower abilities in everyday functional communication and higher scores show better functional communication in everyday life [10].*The standardized Greek version of the Scenario Test-GR [6]*. The Scenario Test-GR is a tool that evaluates functional communication in simulated everyday communication situations. Scoring is completed by the clinician. The Scenario Test-GR consists of 18 items as part of six daily life scenarios (each scenario has 3 questions) using black and white pictures. The score for each item ranges from 0 to 3 for each question. The total score is calculated from the sum of all questions. The scores range from 0 to 54, with a lower score indicating poor functional communication and a higher score indicating better functional communication [9].*The standardized Greek version of the Aphasia Impact Questionnaire-21 (AIQ-21-GR) [41]*. The AIQ-21-GR is a self-reported questionnaire that evaluates the impact of aphasia on the quality of life of PWA. It includes 21 questions and is divided into three domains: participation, communication, and emotional state. Participation includes 7 items, communication 6 items, and emotional state 11 items. Each item has a 5-point rating scale (0–4), with 0 indicating “no problem” and 4 indicating “impossible”. Total scores range from 0 to 84, with lower scores indicating a lower impact of aphasia on quality of life [42].

### 2.8. Testing of Reliability

#### 2.8.1. Test–Retest Reliability

All participants (30 PWA and their 30 respective carers) were engaged in the test–retest reliability phase and were asked to complete the CETI-GR for a second time in a 7–14-day interval after the first administration. 

#### 2.8.2. Inter-Rater Reliability

During the inter-rater reliability phase, the person with aphasia and his/her carer completed the test independently. To assess inter-rater reliability between the two speech and language therapists (EA and MC), the CETI-GR was administered to PWA by both raters (EA and MC). 

### 2.9. Testing of Validity

#### 2.9.1. Convergent Validity

In terms of convergent validity, four hypotheses were formulated:Scores of the CETI-GR will significantly correlate with measures of language, that is, the ASRS of the BDAE-SF. Previous studies have shown a close association between language impairments after stroke and functional communication [6,43].Moderate to high correlation was expected for the Scenario Test-GR since it assesses functional communication, even though the items are rated by the person with aphasia and not the carer [6,43].Scores of the CETI-GR will significantly correlate with measures of the psychosocial domain, that is, the AIQ-21-GR, as there is evidence of a link between low functional communication and poor quality of life [44].The CETI-GR will correlate moderately with the HADS-GR [40]. According to Schumacher et al. [45], there is strong evidence for the importance of assessing mood disorders when language production is impaired in PWA during functional communication assessments.

#### 2.9.2. Content Validity 

The first author (MCha), in consultation with the PPI partner, a young female stroke survivor with chronic mild–moderate anomic aphasia, co-developed a 16-item self-rating questionnaire to assess the importance, comprehensiveness, relevance, and appropriateness of the content of the statements. The questions were created following the Consensus-based Standards for the selection of health Measurement Instruments (COSMIN) guidelines [46]. Different stakeholders completed the questionnaire using a 5-point Likert scale (1 “strongly disagree” to 5 “strongly agree”) to report on content validity. Stakeholders included formal carers (professionals), informal carers (family members), people with aphasia, and people with stroke and no aphasia who did not participate in the psychometric study. To assess the relevance of the 16 items, analyses of the median scores were conducted. An item was accepted as “very relevant” if it received a median score of at least 4. Overall, the results were expected to confirm that the content of the 16 items of the Greek version of the CETI was appropriate.

### 2.10. Criteria for Psychometric Testing

The following criteria were used to test the reliability and validity of the CETI-GR. Generally, a Cronbach’s α > 0.70 indicates good internal consistency [47]. Similar to previous studies measuring the psychometric properties of the CETI, a rounded Cronbach’s α ≥ 0.8 was considered excellent [10,13]. Intraclass correlation coefficients (ICC) ≥ 0.80 indicate good inter-rater reliability of the overall measure [48]; ICCs should be ≥0.75 for good test–retest reliability [48]. Correlational analysis (Spearman’s ρ) was undertaken to test the convergent validity of the measure. Commonly, in psychometric testing, correlations between 0 < ρ < 0.3 or 0.3 < ρ < 0 are considered weak, between 0.4 < ρ < 0.6 or −0.6 < ρ < −0.4 moderate, and ρ > 0.6 or ρ < −0.6 strong [49]. Lastly, the non-parametric Mann–Whitney *t*-test was used to compare the CETI-GR scores of those with aphasia compared to their carers.

### 2.11. Data Analysis

All statistical analyses of the collected data were analyzed with the jamovi (version 1.6) statistics computer software.

## 3. Results

### 3.1. Measures

The descriptive statistics for the scores on the ASRS, CETI-GR, AIQ-21-GR, Scenario Test-GR, and HADS-GR for PWA and their carers are presented in Table 4. Regarding the CETI-GR, PWA reported an average score of 74.49 (sd = 17.92), while their cares reported an average score of 65.71 (sd = 25.60). 

### 3.2. Reliability Analyses

#### 3.2.1. Internal Consistency

The Cronbach α was estimated to calculate the internal consistency of the CETI. The CETI demonstrated excellent consistency (α = 0.95, 95% CI = [0.93, 0.97]). This high consistency remained even when excluding each item independently, decreasing only to 0.94 when dropping Item 10 or Item 12. Item–rest correlations ranged between 0.55 and 0.88, with only three items resulting in correlations below 0.6 (Item 1 = 0.55, Item 11 = 0.57, and Item 13 = 0.57). Further, excellent consistency persisted when testing the CETI responses by PWA only (α = 0.91, 95% CI = [0.86, 0.95]) or by their carers only (α = 0.97, 95% CI = [0.94, 0.98]).

#### 3.2.2. Intraclass Correlations

Intraclass correlations were used to investigate test–retest reliability, as well as the reliability between two examiners (EA and MC), and between PWA and their carers. Test–retest reliability was high (ICC = 0.88, 95% CI = [0.8, 0.93]) and so was reliability between examiners (ICC = 0.93, 95% CI = [0.89, 0.96]). Reliability between PWA and their carers was low (ICC = 0.20, 95% CI = [−0.17, 0.52]). Even though there was low reliability between PWA and their carers, no significant differences were found through a non-parametric Mann–Whitney *t*-test, between the CETI scores of the PWA and their carers (W = 531.5, *p* = 0.23). The individual and average CETI scores from PWA and their carers are illustrated in Figure 2A. Considering the possibility of systematic errors hindering potential differences (Phylactou et al., 2022 [50]) between the groups, we further calculated a difference score by subtracting the CETI scores of PWA by those of their respective carer (Figure 2B). We then conducted a non-parametric Wilcoxon signed-rank one sample *t*-test, testing against a difference of 0. The one sample *t*-test replicated the results of the between groups *t*-test, indicating no significant differences (*V* = 302.5, *p* = 0.15). Of note, a similar pattern was noticed for the AIQ-21-GR scores, where the reliability between PWA and their carers was low (ICC = 0.37, 95% CI = [0.01, 0.64]), but with no statistically significant differences (W = 325, *p* = 0.07). 

### 3.3. Validity Analyses

To test the validity of the CETI-GR, non-parametric correlations (Spearman’s ρ) were used to examine the relationship between the CETI-GR scores and scores on the relevant psychometric tests. With the exemption of the HADS-GR (ρ = −0.322, *p* = 0.08), the CETI-GR correlated significantly with the ASRS (ρ = 0.574, *p* < 0.001), the AIQ-21-GR (*ρ* = −0.461, *p* < 0.001), and the Scenario Test-GR (*ρ* = 0.642, *p* < 0.001). A heatmap of these correlations is shown in Figure 3.

#### Content Validity

The evaluation of the content validity results showed that the 16 items of the CETI-GR received high ratings for importance, comprehensiveness, relevance, and appropriateness, aligning with participants’ communication needs. The content validity questionnaire was completed by *n* = 20 participants who were not part of the psychometric study. This included 4 professional carers, 4 family member carers, 4 PWA, and 4 people with stroke and no aphasia. Overall, the assessment was rated as “very relevant”, indicating an accurate representation of the construct. In total, content validity was given higher scores with an overall median score of 4.5 [Q_25_ = 4, Q_75_ = 5]. No between-group differences were found for the scores of the COSMIN subcategories [46]. High importance scores (median = 4, [Q_25_ = 4, Q_75_ = 4.63], across groups) reveal that items are highly important for assessing communication effectiveness in PWA. High comprehensiveness scores (median = 4, [Q_25_ = 4, Q_75_ = 5], across groups) suggest comprehensive coverage of functional communication. High relevance scores (overall median = 4, [Q_25_ = 4, Q_75_ = 5], across groups) indicate direct applicability to real-life experiences. Also, high appropriateness scores (overall median = 4, [Q_25_ = 4, Q_75_ = 5], across groups) confirm suitability for respondents in terms of language and cultural relevance. Overall, excellent content validity confirms that the CETI-GR effectively captures functional communication concepts as seen in Table 5.

## 4. Discussion

In this study, the Communicative Effectiveness Index (CETI) was adapted into Greek (CETI-GR). Based on a comprehensive psychometric evaluation of the 16 items of the CETI, this study presents findings on its reliability and validity as a functional communication assessment tool for Greek-speaking people with chronic aphasia.

### 4.1. CETI-GR’s Reliability

The CETI-GR demonstrated high internal consistency, test–retest reliability, and inter-rater reliability. The strong evidence of the reliability of the Greek adaptation of the CETI is consistent with the findings for the original English version [10], but also for the language adaptations and validated versions in Danish [12] and Italian [13]. Apart from internal consistency, the psychometric properties of the South African CETI were not reported. The Greek version and the original English version by Lomas et al. [10] used all three aspects of tool reliability, internal consistency, test–retest reliability, and inter-rater reliability, with similar results. 

#### 4.1.1. Internal Consistency

The CETI-GR has excellent internal consistency, indicating high consistency among the items on the scale. Similarly, all the published CETI studies have demonstrated high internal consistency. This means that the scale’s items reliably measure the same underlying construct and have a high correlation, ensuring that they measure related aspects of cross-cultural readiness [51]. Researchers and clinicians can confidently use the CETI-GR to assess the functional communication skills of PWA in everyday settings from the perspective of their carers [51]. Further, excellent consistency continued when testing the CETI-GR responses by either PWA or their carers. Both PWA and their carers showed high internal consistency when responding to the CETI. Similarly, the Danish study [12] found that the CETI had high internal consistency. Penn et al. [11] stated that the CETI appears to be a useful tool in the context of South Africa and is sensitive to the effects of severity and aspects of recovery. Finally, the internal consistency results of the Italian adaptation [13] are consistent with the findings of the current study, supporting the efficacy of the CETI for use in different languages and cultures.

#### 4.1.2. Test–Retest Reliability

The Danish [12] and Italian [13] versions underwent test–retest studies, yielding results akin to those of the current study and the English versions. Specifically, the Italian adaptation of the CETI showed consistent test–retest results for individual items and total scores over a one-week interval [13], similar to this study. In the Danish adaptation [12], although the CETI exhibited good test–retest reliability, the findings revealed lower test–retest reliability compared to the Western Aphasia Battery [14] when the retest was conducted after three and a half months from the initial test. Petersen and colleagues [12] stated that it is important to consider that some patients were undergoing rehabilitation during this time, which might have influenced their language performance and scoring [12].

#### 4.1.3. Inter-Rater Reliability

To test the inter-rater reliability of the CETI-GR, comparisons were made between the ratings of PWA and their carers, as well as between the two raters, EA and MC. The results showed high reliability among all the raters. Nevertheless, there was low reliability between PWA and their carers, but no significant differences were found in the total scores of the CETI-GR. This indicates that while PWA and their carers may not always align in their assessments of specific communication skills, their overall views on communication effectiveness tend to be similar [13]. Individual differences might influence their rating variability in perception, interpretation, or communication abilities [12]. However, this finding emphasizes the importance of considering perspectives from both PWA and their carers in assessing communication abilities and designing interventions or support strategies tailored to the needs and experiences of both parties [12].

Regarding the high reliability between the two speech and language therapists, the results show that the CETI-GR allows for a continuity in therapy when the speech and language therapist is replaced. Therefore, the new speech and language therapist can use the previous results to enhance their therapeutic approach and seamlessly continue administering the CETI-GR to monitor the patient’s progress over time [52]. This consistency ensures that the therapeutic process remains effective and uninterrupted, facilitating better tracking and management of the patient’s communication abilities [52].

#### 4.1.4. Reliability and the Social Context

The reliability results of the two published adaptation studies [12,13] highlight the sensitivity of the measure for assessing functional communication in PWA across different sociocultural groups. Despite differences in linguistic and cultural contexts and the specific aspects of aphasia assessed (verbal and non-verbal communication, reading/writing skills), all studies reported high levels of test–retest reliability for the CETI [12,13]. Several factors may contribute to the findings of consistent reliability across the studies. The standardized testing procedures of the CETI likely ensured consistency in administering and scoring the tool, while clear scoring criteria probably minimized variability between raters or testing occasions. Overall, although linguistic and cultural disparities may impact the adaptation and interpretation of the CETI, the consistently high levels of test–retest reliability observed across the studies indicate that the 16 items can reliably measure functional communication in PWA across diverse contexts.

### 4.2. CETI-GR’s Validity 

The results from the validity analyses revealed significant correlations between the CETI-GR and measures of aphasia severity (ASRS), functional communication (Scenario Test-GR), and quality of life (AIQ-21-GR) but no correlation with measures of depression and anxiety (HADS-GR).

#### 4.2.1. Convergent Validity

##### Correlation of the CETI-GR with Aphasia Severity

The first hypothesis proposed a correlation between aphasia severity and functional communication. In line with prior studies by Lomas et al. [10], Pedersen et al. [12], and Moretta et al. [13], a comparative analysis between the scores obtained from the CETI-GR and those from the Aphasia Severity Rating Scale (ASRS) [39] was conducted. The correlation between ASRS and the CETI-GR measures was strong, similar to the correlations found in the aforementioned studies. Specifically, the original study by Lomas et al. [10] revealed that the CETI was highly correlated with the three measures of aphasia performance of the Western Aphasia Battery [14]. The two subtests of the Western Aphasia Battery included the Aphasia Quotient, which is similar to the ASRS, and the Speaking and Understanding subscales. Similarly, Pedersen et al. [12] found high correlations between the Danish version of the CETI and the Western Aphasia Battery [14], while in the Italian CETI adaptation, Pedersen et al. [12] found strong correlations with the Brief Exam of Language Impairment-II [15]. 

Moreover, the correlation between the CETI-GR and ASRS was significant, considering the diversity in aphasia severity. In the Greek sample, participants with more severe language difficulties (1 and 2 on the ASRS) scored lower on questions evaluating everyday social and communication situations, for example, item 7: “Having a one-to-one conversation” and item 6: “Having coffee-time, visits, and conversations with friends and neighbors”. Similar findings were evident in the studies by Lomas et al. [10], Penn et al. [11], and Moretta et al. [13], who stated that the CETI demonstrates applicability across a wide spectrum of language impairments, serving as a reliable tool for evaluating residual communication skills post stroke. Additionally, Lomas et al. [10] proposed that the CETI can be applied to PWA with varying degrees of language severity. Finally, in the South African adaptation, Penn et al. [11], concluded that the CETI can effectively distinguish between mild and severe cases of anomia, a characteristic symptom of aphasia. 

##### Correlation of the CETI-GR with the Scenario Test

The second hypothesis was for a high correlation between the CETI-GR and the Scenario Test-GR, a test of functional communication in PWA. As predicted, there was a strong correlation between the scores of the CETI-GR and the Scenario Test-GR.

The CETI-GR demonstrated a higher correlation with the Scenario Test-GR compared to the ASRS, revealing that PWA showed better communication abilities, irrespective of the severity of the language impairment. The outcome is justified by the distinct focus of the CETI-GR, which evaluates functional communication abilities, unlike the ASRS, which assesses language output at the impairment level. Additionally, the CETI correlated with other measures of functional communication, as seen in the original study by Lomas et al. [10] and the Danish adaptation [12]. Specifically, the original version of the CETI [10] correlated with the Speech Questionnaire of the Western Aphasia Battery [14], which includes items on functional communication abilities [53]. Similarly, the Danish version of the CETI [12] correlated with the Porch Index of Communicative Ability (PICA) [16], a measure of functional communication in everyday settings.

##### Correlation of the CETI-GR with the Aphasia Impact Questionnaire 

The third hypothesis proposed a significant correlation between the CETI-GR and the AIQ-21-GR, as there is evidence for a link between poor functional communication skills and reduced quality of life. Interestingly for this study, the CETI-GR showed a high correlation with the AIQ-21-GR [41]. Several studies have reported the importance of social interaction, life participation, and friendships, which are negatively affected because of impaired communication abilities [54]. Previous studies have also reported a close association between quality of life and functional communication, and how the psychosocial domain is related to functional communication abilities [55]. 

The results of the Italian adaptation [13] are in line with the results of this study. Specifically, the Italian version of the CETI showed significant correlations between aphasia impact on daily life and functional communication abilities when comparing the CETI with the Activities of Daily Living (ADL) scale [17] and the Activities of Daily Living (IADL) scale [18]. On the contrary, Pedersen et al. [12] used the Barthel Index [19] and the Frenchay Activity Index [20] to explore a possible correlation between functional communication and its impact on activities of daily living but they did not find any significant correlation between the three tools. According to Pedersen et al. [12], this was evident because the CETI primarily assesses communication abilities rather than encompassing overall daily activity performance alone. 

To ensure the accuracy of our findings, carers also completed the AIQ-21-GR [41] for PWA. The results revealed a distinction in the scoring of the CETI-GR and the AIQ-21-GR between carers and PWA. This result stands out for the Greek study, as previous validation studies had not undertaken such a comparison. As evidenced by the results of this study, carers rated the functional communication abilities of PWA and the impact of aphasia on their everyday life lower than the self-assessments provided by the PWA. One potential explanation could be that carers may have a heightened awareness of the communication challenges faced by PWA, considering the broader impact of aphasia on their ability to engage in social interactions and carry out daily activities [27,56]. This heightened awareness may lead carers to evaluate functional communication differently, potentially resulting in variations in scoring.

In contrast, the discrepancy in scoring as perceived by PWA may be ascribed to a multitude of contributing factors. Firstly, PWA may have a more intimate understanding of their communication abilities, including their strengths, weaknesses, and compensatory strategies, leading to a more positive self-assessment [57]. Additionally, PWA may focus on successful communication instances or improvements over time, that could contribute to a more optimistic perception of their abilities [58]. Also, PWA may prioritize different aspects of communication than their carers, leading to discrepancies in perceived skill levels [59]. Lastly, PWA may experience frustration or emotional distress related to their communication difficulties [60], which could influence their self-assessment differently than the perceptions of their carers.

##### Correlation of the CETI-GR with Depression and Anxiety 

The final hypothesis proposed a moderate correlation between functional communication skills, depression, and anxiety [3]. This was tested by comparing the CETI-GR with the Greek version of the HADS [40]. However, this correlation was not significant, indicating limited evidence of an association between communication effectiveness and levels of anxiety and depression. One possible reason for this is that the HADS, designed to assess mood disorders in individuals with medical conditions, may not be suitable for people with communication impairments like aphasia. Additionally, its structure may not accommodate the needs of chronic patients, potentially limiting its effectiveness in this population. Similar findings were observed in the Danish adaptation of the CETI, where Pedersen et al. [12] compared the CETI with the Non-Verbal Index of Depression [22], revealing no correlation between the two tools. Pedersen et al. [12] attributed this to the CETI’s focus on evaluating patients’ communication in everyday situations while disregarding their emotions or the impact of the communication impairment on mood in such contexts. Further, in the Italian adaptation, Moretta et al. [13] aimed to investigate a correlation between the CETI and depression using the Aphasic Depression Rating Scale (ADRS) [21], again, with no correlation between the ADRS and the CETI. 

#### 4.2.2. Content Validity 

As mentioned in the Section 2, the adaptation process of the CETI into Greek involved a content validity study of the 16 items, ensuring linguistic and contextual appropriateness for Greek-speaking populations in Cyprus. The content validity study demonstrated high ratings around the importance, comprehensiveness, relevance, and appropriateness of the CETI-GR items. The content validity assessment was deemed as “very relevant”, which means that the content of the 16 items accurately represents the construct it is intended to measure [46]. Specifically, a high score of importance means that most of the stakeholders find the items to be highly important for assessing communication effectiveness in people with aphasia. Also, the very high score in comprehensiveness means that most items collectively provide a comprehensive view of functional communication without omitting critical components. In terms of relevance, the results show that the CETI-GR is directly applicable and meaningful to the target population reflecting real-life experiences, situations, or behaviors that are pertinent to the construct of the CETI-GR. Finally, the high scores regarding the appropriateness of the items confirm that the 16 items are suitable for PWA and their carers in terms of language, context, and cultural relevance, ensuring that respondents understand and reply to the items effectively. In summary, excellent content validity indicates that the CETI-GR is a well-designed tool that captures functional communication concepts for Greek-speaking people.

### 4.3. The Use of the Modified Index 

Pedersen et al. [12] observed that during the completion of the Danish CETI, raters often placed their crosses midway between the endpoints of the visual analog scale lines when intending to assign a score of 0 or 100. As a result, they recommended that future researchers supervise participants during the rating process to prevent such misunderstandings. In this study, their suggestion was adhered to, ensuring that all participants were supervised by the two speech and language therapists. This proactive measure significantly enhanced the precision and reliability of the data. Also, to support the precise marking of the rating scale of the Index, smiley faces were added at the beginning, middle, and end of the Index (see Figure 1) to represent the main indicators for “not at all able” to “as able as before”. Based on the observations and feedback of the speech therapists (AE and MC) this strategy also helped PWA in scoring the Index more accurately, rather than marking the scale randomly. Nevertheless, there are no data collected on how much the use of smiley faces helped PWA to score better. This could be an area of further research with a comparative analysis using smiley faces and a study with the standard format to improve the adaptation and the further development of the CETI-GR questionnaire. 

### 4.4. Clinical Implications

The CETI-GR was shown to be useful in assessing the effects of aphasia on the functional communication abilities of PWA in daily life. One potential benefit of utilizing the CETI-GR in clinical practice is its ability to allow carers to measure language use in situations when a clinician’s direct language-based assessments could not be provided. Moreover, employing this objective and quantitative metric for functional communication can yield more precise measurements and effectively demonstrate progression in the time course of aphasia rehabilitation.

The CETI-GR can help clinicians follow up with patients at home and develop treatments tailored to the “ecologic” difficulties of patients, thereby improving their quality of life [13]. Moreover, the CETI-GR relies on assessments completed by the carers, providing clinicians with organized first-hand evidence of communication performance and involving carers in the assessment process [12]. This involvement can motivate carers and orient them to the range of possible communication behaviors the person with aphasia may exhibit [13]. Enhanced participation from those who live with or spend significant time with the person with aphasia may result in more functional goal setting, as clinicians typically see the person with aphasia only a few hours per week [61]. The significance of adapting the CETI into Greek cannot be overstated. The CETI-GR is now available to Greek-speaking aphasia clinicians working in the chronic phase, an often understated phase of aphasia rehabilitation [52]. Finally, the availability of the CETI-GR opens doors for tailored interventions and therapies that address the unique communicative challenges faced by Greek-speaking PWA, ultimately enhancing their quality of life and rehabilitation outcomes [62].

### 4.5. Limitations of the Study

It is important to acknowledge certain limitations of the present study when interpreting the results. Despite the limited number of PWA in the sample, the multicentric design of the research significantly enhances the generalizability of the findings. By involving multiple centers and all major cities around the country, researchers captured a broader representation of the population affected by aphasia in Cyprus, increasing the applicability of the results to diverse settings [61]. However, only individuals with chronic post-stroke aphasia were recruited, whereas previous studies included PWA in both the chronic and the acute phases [10,11,13]. Also, the wide variation in the months post stroke, ranging from 6 to 180 months, posed a challenge in drawing specific conclusions about the functional communication performance of PWA in the various stages of stroke recovery [62]. This variability in timing could have influenced the efficacy of the functional communication skills or the natural progression of stroke and aphasia recovery, making it difficult to pinpoint precise correlations between the therapeutic outcomes and the time since the stroke [62]. Additionally, this wide aphasia chronicity timeline may have affected how PWA rated their communication abilities. According to Nichol et al. [63], individuals entering the chronic phase (6 months post stroke) might not have been able to adjust and self-manage their communication difficulties and psychosocial challenges after the stroke. Therefore, participants might have rated themselves lower than those who have achieved self-management and developed compensatory strategies for their communication impairments at later stages, e.g., more than 2 years post stroke. Finally, none of the participants experienced very severe aphasia, as the ASRS scores ranged from mild to moderate/severe aphasia, with the majority (50%) experiencing mild aphasia.

### 4.6. Future Directions

The CETI-GR could be enhanced by involving PWA and their carers as research partners during the test adaptation process. This could add practical value to the test by engaging all stakeholders in the co-design and co-production of the items to improve the feasibility, usability, and relevance of the tool [23]. This active involvement may also help provide specific examples that need to be modified based on knowledge and lived experiences and offer suggestions for the improvement of the content. Therefore, a potential future direction is to review or add new test items that could focus on supporting communication skills in various situations not covered in the original CETI. For example, additional items related to the functional communication abilities of PWA could include daily situations such as discussing (a) banking services, (b) healthcare environments and doctor’s visits (e.g., at the hospital and during medical tests), (c) accessing digital technologies (computers, smartphones, and the internet), and (d) medical or other emergencies. These new topics would enable therapists to facilitate context-specific communication interactions and design relevant intervention approaches.

## 5. Conclusions

The findings support the reliability and validity of the CETI-GR as a valuable assessment tool for clinicians working with Greek-speaking individuals with chronic aphasia. It is crucial to involve carers in the assessment of functional communication of people with aphasia to gain valuable insights into daily communication challenges, and for clinicians to gain a comprehensive understanding of communication abilities in real-life contexts. Overall, the CETI-GR can be used to inform personalized intervention plans, facilitate collaborative goal setting, and ultimately improve the quality of life for people living with aphasia and their carers in Greek-speaking communities.

## Figures and Tables

**Figure 1 brainsci-14-00689-f001:**
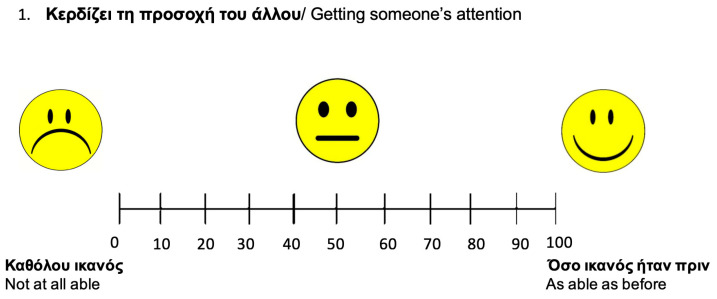
An example of the visual analog scale for Item 1 with smiley faces.

**Figure 2 brainsci-14-00689-f002:**
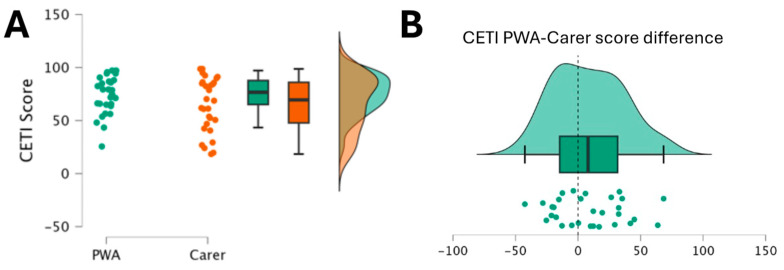
(**A**) Individual and averaged CETI-GR scores for PWA (green) and their carers (orange). (**B**) Difference in CETI-GR scores between PWA and their respective carers. Notes: PWA, people with aphasia; CETI-GR, Greek version of the Communication Effectiveness Index.

**Figure 3 brainsci-14-00689-f003:**
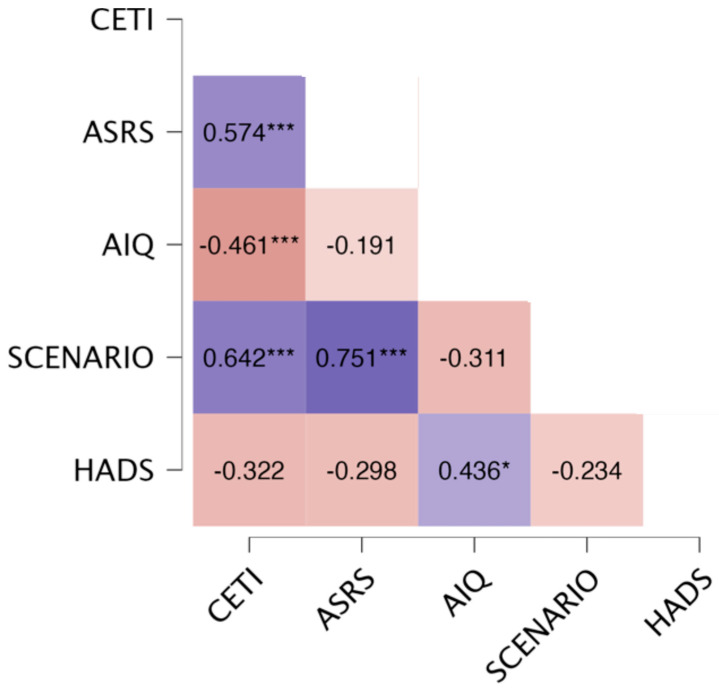
A heatmap reporting on the correlations between the CETI-GR, the ASRS, the AIQ-21-GR, the Scenario Test-GR, and the HADS-GR. Notes: *** *p* ≤ 0.01, * *p* ≤ 0.05; ASRS, Aphasia Severity Rating Scale; CETI, the Communication Effectiveness Index; AIQ, the Aphasia Impact Questionnaire; HADS, Hospital Anxiety and Depression Scale.

**Table 1 brainsci-14-00689-t001:** Participant and adaptation information from the published studies on the CETI.

CETI Version	English	South African	Danish	Italian
Participants (*n*)	33	56	68	136
PWA (*n*)	22	22	68	68
Stroke no aphasia (*n*)	-	6	-	-
Carers (*n*)	11	28	-	68
Psychometric Properties				
*Reliability*				
Internal consistency	+	+	+	+
Test–retest	+	+	+	+
Inter-rater	+	-	+	+
*Validity*				
Construct	+	-	+	+
Convergent	+	-	+	+
Tools				
*Language*				
Western Aphasia Battery (WAB) [14]	+	-	+	-
Brief Exam of Language Impairment-II [15]	-	-	-	+
*Functional Communication*				
Speech Questionnaire (SQ) of the WAB [14]	+	-	+	-
Porch Index of Communicative Ability (PICA) [16]	-	-	+	-
*Activities of Daily Living*				
Activities of Daily Living scale (ADL) [17]	-	-	-	+
Activities of Daily Living scale (IADL) [18]	-	-	-	+
Barthel Index (BI) [19]	-	-	+	-
Frenchay Activity Index (FAI) [20]	-	-	+	-
*Depression*				
Aphasic Depression Rating Scale (ADRS) [21]	-	-	-	+
Non-Verbal Index of Depression (NID) [22]	-	-	+	-
Beck Depression Inventory (BDI) [23,24]	-	-	+	-

**Table 2 brainsci-14-00689-t002:** Sample size of the CETI-GR compared to other adaptations.

Country	Population	Incidence Estimate	CETI Sample Size (*n*) + [Study]
Canada	39,034,588	50,000 strokes/year	*n* = 33 [10] 22 PWA + 11 carers
South Africa	54,956,900	120,000 strokes/year	*n* = 56 [11] 28 PWA + 28 carers
Denmark	5,368,854	5297 strokes/year	*n* = 68 PWA [12]
Italy	60,665,625	73,116 strokes/year	*n* = 136 [13] 68 PWA + 68 carers
Cyprus	803,147	564 strokes/year	*n* = 60 (current study) 30 PWA + 30 carers

Note: PWA, people with aphasia.

**Table 3 brainsci-14-00689-t003:** Participant demographics, level of education, marital and socioeconomic status.

*Characteristics*	*People with Aphasia* *(n = 30)*	*Carers* *(n = 30)*
*Gender*		
Male	16 (47%)	7 (77%)
Female	14 (53%)	23 (23%)
*Age*		
Mean (sd)	67.67 (10.71)	47.4 (16.33)
Min–Max	36–89	26–78
*Stroke Type*		
Ischemic	15 (50%)	N/A
Hemorrhagic	12 (40%)	N/A
Other	3 (10%)	N/A
*Lesion Location*		
Left	15 (50%)	N/A
Right	15 (50%)	N/A
*Hemiplegia*		
Left	8 (27%)	N/A
Right	10 (33%)	N/A
None	12 (40%)	N/A
*Months Post Stroke Diagnosis*		
Mean (sd)	44.07 (48.81)	N/A
Min–Max	6–180	N/A
*Completed Education*		
Primary	4 (13%)	0
Secondary	19 (63%)	7 (23%)
College	0	4 (13%)
Bachelor’s	5 (17%)	8 (27%)
Master’s	2 (7%)	9 (30%)
PhD	0	2 (7%)
*Marital Status*		
Married	13 (43%)	16 (53%)
Single	5 (17%)	11 (37%)
Divorced	4 (13%)	1 (3%)
Widowed	8 (23%)	2 (7%)
*Socioeconomic Status Based on Former or Current Occupation*		
Higher managerial, e.g., chief executive officer	3 (10%)	6 (20%)
Intermediate occupation, e.g., clerical, sales, service	5 (16%)	16 (53%)
Manual occupation, e.g., painter, builder	14 (47%)	6 (20%)
Unemployed	8 (27%)	2 (7%)

**Table 4 brainsci-14-00689-t004:** Group means on the Aphasia Severity Rating Scale (ASRS), the CETI-GR, the AIQ-21-GR, the Scenario Test-GR, and the HADS for PWA and their carers.

	Group	Mean	sd	Minimum	Maximum
*ASRS*	PWA	3.3	1.58	1	5
	Carer	-	-	-	-
*CETI-GR*	PWA	74.49	17.92	25.63	97.19
	Carer	65.71	25.6	18.44	98.75
*AIQ-21-GR*	PWA	23.43	14	0	47
	Carer	30.63	13.62	6	52
*Scenario Test-GR*	PWA	38.2	19.05	0	54
	Carer	-	-	-	-
*HADS-GR*	PWA	11.67	6.39	3	25
	Carer	-	-	-	-

Notes: ASRS, Aphasia Severity Rating Scale; CETI-GR, Greek version of the Communication Effectiveness Index; AIQ-21-GR, Greek version of the Aphasia Impact Questionnaire; Scenario Test-GR, Greek version of the Scenario Test; HADS-GR, Greek version of the Hospital Anxiety and Depression Scale.

**Table 5 brainsci-14-00689-t005:** Median scores of each group for each subcategory of the content validity questionnaire.

			Percentiles	
	Group	Median	25th	75th
* Importance *	PWA	4.5	4.38	4.63
	SNA	4	4	4.25
	Professional Carers	4	4	4.25
	Family Carers	4	4	4.25
* Comprehensiveness *	PWA	5	5	5
	SNA	4	4	4.25
	Professional Carers	4	4	4.25
	Family Carers	4	4	4.25
* Relevance *	PWA	5	4.75	5
	SNA	4	4	4.25
	Professional Carers	4	4	4.25
	Family Carers	4	4	4.25
* Appropriateness *	PWA	5	4.75	5
	SNA	4	4	4.25
	Professional Carers	4	4	4.25
	Family Carers	4	4	4.25

Notes: PWA, people with aphasia; SNS, stroke no aphasia.

## Data Availability

The data generated during the current study and supporting the conclusions of this article are publicly available in the manuscript. Any further data queries and requests should be submitted to the corresponding author, Marina Charalambous PhD, for consideration.

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
