# Peer review of "The Adaptation of the Communicative Effectiveness Index (CETI) into Greek: A Reliability and Validity Study"

_brainsci, 2024, doi:10.3390/brainsci14070689_

Round 1
Reviewer 1 Report
Comments and Suggestions for Authors
Review
“The Adaptation of the Communicative Effectiveness Index
(CETI) into Greek: A Reliability & Validity Study”
General assessment:
- This article explores the psychometric properties of the so-called “Communicative Effectiveness Index” (CETI). The CETI is a scale used to measure the communicative abilities of individuals affected by speech and language impairments such as aphasia in order to enhance rehabilitation. It is designed to provide a way to assess how effectively a person can communicate in various everyday situations from the caregiver’s perspective. The scale was translated into and adapted to Greek; one of the primary objectives of the study was to ensure that the assessments provided by the CETI were reliable, valid, and appropriate for the population being studied (in the investigation presented here, 30 people with aphasia and their caregivers). The main result of the study is that the Greek version of the CETI (CETI-GR) reveals a dependable tool for evaluating functional communication in individuals with chronic aphasia.
- The paper is certainly relevant to the scope of the special issue “The Impact of Language(s), Social Environment and Culture on Brain Development and Function”. The paper reads very well, the style is impeccable, the language is excellent. I could not even find a typo, which means that the text has been appropriately revised.
- I certainly support the publication of the paper in BrSci. There are, however, a couple of points that the authors should consider before submitting the final version of the paper (see below).
For these reasons, my overall recommendation is:
ACCEPT AFTER MINOR REVISION
Content:
- An aspect that should be addressed in more detail in the text concerns the methods introduced in Section 2 (and already mentioned in the abstract): it is not clear to me why the questionnaire is back-translated into English, at least for two reasons: 1) The study presented here involves Greek-speaking people with aphasia; 2) in my understanding of the notion of back-translation, a text is translated into its original language by a translator who is not familiar with the content of the original. In the text of the paper, however, it is claimed that the translator is one of the authors. It would therefore be surprising if the translation had not resulted in a high correspondence to the original source. Please explain this more in detail.
- The caregivers, who are not experts (in the academic sense) in the evaluation of the variables investigated here, are certainly fundamental actors in the context of the CETI, since they provide the feedback that is necessary to test the validity of the questionnaire. One potential issue concerns the comparability of the caregivers’ assessments: how do the authors know what kind of attitude the caregivers have when evaluating the different items (e.g., if they are able to evaluate the competences w.r.t. the previous stage or according to objective criteria)? Please elaborate on it.
- I am surprised to see that the paper does not address the type(s) of Aphasia involved in the study. What kind of Aphasia does the test persons have? Which areas of the brain are affected? This makes a huge difference w.r.t. the linguistic output: say, Broca’s Aphasia and Wernicke’s Aphasia dramatically differ in terms of (a.m.o. aspects) fluency, speech effort, (presence/absence and incidence of) agrammatism, paraphasias, comprehension, etc. … Is this test appropriate for the evaluation of all types of Aphasia? If yes, how? And how did the authors differentiate the results/assessments coming from the caregivers in this respect? I do not think that the authors did not consider this, but it should be discussed much more in detail.
- It would be desirable to see more original material from the CETI apart from the general items (“Getting someone’s attention”, etc.): what kind of stimuli are involved? How are the results collected exactly?
Formal aspects:
- p. 1 and elsewhere: all abbreviations should be spelled out upon their first occurrence in the text. It is certainly clear what, e.g., “PWA” means, but for the sake of formality, this should be made explicit. In other cases, e.g. with the acronym “PPI”, this is in fact done.
Author Response
Please find our responses to all questions/queries in the attached PDF under Reviewer 1.

Reviewer 2 Report
Comments and Suggestions for Authors
The paper is well-structured and the language is clear. In my view, there are no major deviations from the standard procedures used in in the adaptation of scales. I have, though, a few questions regarding some choices made by the authors.
1. The reasoning behind sample size justification is not totally clear: are the authors considering a “minimal” proportion (which, by the way?) between (a) sample size and population, (b) sample size and number of strokes, or something else? Moreover, shouldn’t the number of items in the scale (16) matter too? And the statistical tests used (a priori power analysis)?
2. It is not clear either why the level of depression and anxiety is measured for convergent validity. Is there empirical evidence that lack of communication effectiveness overlaps with mood disorders?
3. The poor correlation between carer and patient ratings was toned down by the authors with the argument that average scores were not significantly different. Could not this happen by mere chance, i.e., dyads were never aligned, sometimes carers provided higher scores, sometimes lower, and amidst the “noise”, group differences vanished?
Comments on the Quality of English LanguageNo major issues detected
Author Response
Please find our responses to all questions/queries in the attached PDF under Reviewer 2.

Reviewer 3 Report
Comments and Suggestions for Authors
Dear Authors,
The manuscript contains research material of scientific novelty, but there are a number of shortcomings that require correction.
- Line 205 The percentage of men does not need to be indicated, as it follows from the indicated percentage of women. This can be replaced by a reference to Table 3.
- Line 206 The data duplicates the table. Replace it with a reference to Table 3.
- Line 210 The proportion of males need not be indicated. See the comment above. Replace it with a reference to Table 3.
- In figure 3, the upper left half of the matrix can be omitted as it duplicates the lower left half.
Author Response
Please find our responses to all questions/queries in the attached PDF under Reviewer 3.

Reviewer 4 Report
Comments and Suggestions for Authors
The article aims to adapt the CETI methodology for the Greek population. The topic developed in this article is very important and relevant, primarily from an applied perspective. The gap of the study lies in the absence of an adapted and validated version of the CETI methodology for the Greek population. Compared with other published material, this study adds to the subject area by providing information about the potential application of the CETI methodology for a new audience. The authors also provide valuable data on the results obtained from a new cultural sample. I have no questions regarding the research methodology. The references are appropriate.
I have a few comments regarding the article revision. The first paragraph cannot include multiple references to just one source. For example, in the section "1.2. The Communication Effectiveness Index (CETI)," the authors cite only one source numbered 10. The authors describe the procedure for preparing the methodology well. They performed the minimally necessary actions that are standard for the procedure of translating and adapting methodologies.
In conclusion, I want to note the high-quality and substantive illustrations prepared by the authors.
Author Response
Please find our responses to all questions/queries in the attached PDF under Reviewer 4.
